# Computational Modeling to Identify Drugs Targeting Metastatic Castration-Resistant Prostate Cancer Characterized by Heightened Glycolysis

**DOI:** 10.3390/ph17050569

**Published:** 2024-04-29

**Authors:** Mei-Chi Su, Adam M. Lee, Weijie Zhang, Danielle Maeser, Robert F. Gruener, Yibin Deng, R. Stephanie Huang

**Affiliations:** 1Department of Experimental and Clinical Pharmacology, College of Pharmacy, University of Minnesota, Minneapolis, MN 55455, USA; su000055@umn.edu (M.-C.S.); leeam@umn.edu (A.M.L.); rgruener@umn.edu (R.F.G.); 2Bioinformatics and Computational Biology, University of Minnesota, Minneapolis, MN 55455, USA; zhan6385@umn.edu (W.Z.); maese005@umn.edu (D.M.); 3Department of Urology, Masonic Cancer Center, University of Minnesota Medical School, Minneapolis, MN 55455, USA; dengx103@umn.edu

**Keywords:** metastatic castration-resistant prostate cancer, glycolysis, oxidative phosphorylation, cancer metabolism reprogramming, drug repurposing

## Abstract

Metastatic castration-resistant prostate cancer (mCRPC) remains a deadly disease due to a lack of efficacious treatments. The reprogramming of cancer metabolism toward elevated glycolysis is a hallmark of mCRPC. Our goal is to identify therapeutics specifically associated with high glycolysis. Here, we established a computational framework to identify new pharmacological agents for mCRPC with heightened glycolysis activity under a tumor microenvironment, followed by in vitro validation. First, using our established computational tool, OncoPredict, we imputed the likelihood of drug responses to approximately 1900 agents in each mCRPC tumor from two large clinical patient cohorts. We selected drugs with predicted sensitivity highly correlated with glycolysis scores. In total, 77 drugs predicted to be more sensitive in high glycolysis mCRPC tumors were identified. These drugs represent diverse mechanisms of action. Three of the candidates, ivermectin, CNF2024, and P276-00, were selected for subsequent vitro validation based on the highest measured drug responses associated with glycolysis/OXPHOS in pan-cancer cell lines. By decreasing the input glucose level in culture media to mimic the mCRPC tumor microenvironments, we induced a high-glycolysis condition in PC3 cells and validated the projected higher sensitivity of all three drugs under this condition (*p* < 0.0001 for all drugs). For biomarker discovery, ivermectin and P276-00 were predicted to be more sensitive to mCRPC tumors with low androgen receptor activities and high glycolysis activities (AR(low)Gly(high)). In addition, we integrated a protein–protein interaction network and topological methods to identify biomarkers for these drug candidates. *EEF1B2* and *CCNA2* were identified as key biomarkers for ivermectin and CNF2024, respectively, through multiple independent biomarker nomination pipelines. In conclusion, this study offers new efficacious therapeutics beyond traditional androgen-deprivation therapies by precisely targeting mCRPC with high glycolysis.

## 1. Introduction

Prostate cancer (PC) is the second leading cause of cancer deaths in men in the United States [1]. Many late-stage patients develop castration-resistant prostate cancer (CRPC). CRPC is marked by a serum testosterone level below 1.7 nmol/L and the existence of disease progression [2]. Although androgen receptor-signaling inhibitors (ARSI), such as abiraterone [3,4] and enzalutamide [5,6], have significantly improved clinical outcomes of CRPC patients, nearly all CRPC patients inevitably develop further resistance to ARSI [7]. However, therapeutic options are very limited for ARSI-resistant CRPC, highlighting an unmet need for CRPC patients.

Metabolic reprogramming constitutes a hallmark of cancer [8]. In PC, metabolic reprogramming is significantly associated with disease progression to CRPC [9,10]. Previous studies have shown that patients with CRPC exhibited higher levels of glycolysis and oxidative phosphorylation (OXPHOS) compared with those having primary PC tumors [9]. Also, enzalutamide resistance has been linked to increased glycolysis [11], highlighting metabolic adaptation as a versatile survival strategy of cancer cells. This suggests a potential drug resistance mechanism through metabolism reprogramming. Additionally, a clinical study revealed that mCRPC patients with high glycolysis had worse survival rates compared to those with low glycolysis [12]. Furthermore, the study found that AR negative combined with high glycolysis was related to a higher risk of death, underscoring the prognostic value of metabolic profiling in therapeutic decision making [12]. Taken together, the presence of altered metabolism in CRPC progression and treatment resistance may introduce a new target for designing therapies against this deadly disease. However, finding drugs to efficiently inhibit high glycolysis tumor growth is challenging due to metabolic plasticity, including an interplay between glycolysis and OXPHOS [13,14]. For instance, strategies to target a single metabolic enzyme often lead to therapeutic resistance [13], while a drug (e.g., a glutamine antagonist, JHU083) that is primarily designed to target glutamine metabolism but also impacts multiple metabolic targets has shown reasonable antitumor effects due to the weakened ability of cancer cells to rewire metabolic pathways [15], indicating the importance of a holistic approach in targeting cancer metabolism. Therefore, targeting mCRPC with either high glycolysis activities (scores) or both with high glycolysis/OXPHOS activities (scores), instead of targeting a single enzyme, could potentially achieve better disease control and preclude drug resistance.

Traditional drug discovery and development are time-consuming and costly. The preclinical discovery phase, including disease target identification and validation, compound screening assay development, lead compound identification, and optimization, often takes years before getting to the human study stage [16,17]. Drug repurposing, on the other hand, is an appealing alternative way to identify new indications from approved or investigational drugs in a shorter period [18]. In fact, drug-repurposing approaches have contributed to about 30% of newly FDA-approved drugs [19]. Many successful drug repositioning cases have shown promising value in the treatment of various cancers. For instance, thalidomide, originally indicated for morning sickness in pregnant women, was repositioned for combination treatment with dexamethasone for multiple myeloma [20]. Aspirin and raloxifene, originally indicated for pain relief and osteoporosis prevention [21], respectively, have found new roles in reducing colorectal cancer risk [18,22] and breast cancer incidence [23], demonstrating the transformative potential of repurposing in oncology. In recent years, machine learning approaches have been extensively applied in various fields of drug discovery [24] (including for drug repurposing). This data-driven predictive analytics enables us to capture and explore the relationship between predictive variables and target variables. For example, models have been constructed between predictive variables ranging from omics data (e.g., DNA, RNA, protein) [25] to clinical data [26] and target variables like drug responses [25], disease progression [27,28], and side effects [29]. Among these approaches, the availability of substantial in vitro measured drug response data from a large collection of cancer cell lines (CCLs) and multi-omics data has enabled the integration of machine learning tools in drug discovery efforts, facilitating a more efficient and targeted approach to drug development. We have previously reported a computational method, OncoPredict [30], based on a machine learning framework to facilitate drug repurposing. Briefly, this approach first learns drug–gene relationships from measured CCL drug responses and CCL transcriptome profiles. The learned relationship is subsequently utilized to project patient responses to a large collection of drugs using patient tumor gene expression profiles. Predicted patient sensitivity to various drugs can then be examined with phenotypic data to propose candidate compounds addressing different therapeutic needs. This pipeline has been successfully applied to a number of clinical datasets to accurately project patient responses to drugs [31].

Beyond drug nomination, many biomarker-directed treatments have clearly demonstrated a significant clinical outcome improvement in patients with mCRPC. For example, patients with mCRPC carrying germline and somatic mutations in homologous recombination repair genes (e.g., *BRCA1*, *BRCA2*, *ATM*) show improved outcomes when treated with poly-ADP ribose polymerase (PARP) inhibitors [32] and platinum-based chemotherapy [33,34]. Furthermore, mCRPC patients who exhibit an androgen receptor splice variant (AR-V7) in their circulating tumor cells showed better overall survival outcomes when treated with taxanes compared to ARSI treatments [35]. This evidence supports the critical role of biomarker-based treatment approaches in optimizing therapeutic outcomes for mCRPC patients.

Here, we aim to employ such a machine learning method to identify and repurpose drugs with potent efficacy in mCRPC characterized by heightened glycolysis activity. We applied the computational pipeline to two independent mCRPC patient cohorts to nominate drugs of interest and subsequently validate them in PC cell lines. These drugs, once clinically validated, can improve the treatment outcomes of deadly diseases like mCRPC. To facilitate the optimal application of the newly nominated drug(s), we also carried out biomarker discovery to refine patient selection criteria for treatment, emphasizing the move toward personalized medicine in oncology.

## 2. Results

### 2.1. Identifying Candidate Drugs for mCRPC with High Glycolysis

We built drug response prediction models using our OncoPredict (version 0.2) pipeline [30]. For the training data, we employed two primary sources of CCL drug sensitivity screens: CTRPv2 and PRISM. CTRPv2 contains measured responses to 545 drugs, covering about 375 mechanisms of action, in approximately 829 CCLs. In comparison, PRISM contains measured responses to 1419 drugs, covering about 464 mechanisms of action, in about 481 CCLs, notably including a wide array of non-oncology drugs, thereby broadening the potential for repurposing existing medicines for oncology applications. We imputed patient tumor drug response scores in the two independent clinical studies of CRPC patients (SU2C/EC and SU2C/WC). The imputed drug response scores were correlated with calculated tumor glycolysis scores to identify the drugs of interest (Drug_HG_), which are drugs whose imputed sensitivity scores are significantly and negatively associated with tumor glycolysis scores. Similarly, dual-effect drugs (Drug_DE_) were identified as those drugs whose imputed sensitivity scores were significantly and negatively associated with both glycolysis and OXPHOS scores. Only drugs identified in both SU2C/EC and SU2C/WC studies were considered potential drug candidates for subsequent analyses (Figure 1 and Appendix A).

A summary of the findings of drug nomination can be found in Table 1. Specifically, for Drug_HG_, there are 4 and 54 drug candidates nominated in both SU2C/EC and SU2C/WC clinical studies using either CTRPv2 or PRISM drug screens as training datasets, respectively. For Drug_DE_, there are 0 and 19 drug candidates nominated in both clinical studies from these two sources of training datasets. We narrowed down the final candidate list by focusing on drugs that share the same mechanism of action (MOA). Across Drug_HG,_ nine drugs show an over-representation of one of these MOAs: aurora kinase inhibitors, CDK inhibitors, EGFR inhibitors, HSP inhibitors, mTOR inhibitors, and topoisomerase inhibitors. Indeed, some of these mechanisms have been reported to be related to cancer glycolysis. For instance, EGFR signaling and CDKs have been associated with elevated aerobic glycolysis [36]. Also, aurora kinase A inhibitors have demonstrated glycolysis suppression in glioblastoma [37]. Similarly, HSP90 inhibitors exhibited glycolysis reduction in advanced PC [38]. Appendix A provides a full list of the drug candidates, including Drug_HG_ and Drug_DE_.

To further narrow down our candidate drug list, a filtering strategy depicted in Figure 1 was employed. First, the candidate drugs were filtered based on their development status. Only the 22 drugs that were either already approved by the FDA or were in clinical trial stages were carried forward (Appendix A). Then, we checked consistency across datasets. A total of 17 of the 22 drugs were consistently predicted to be efficacious in both high-glycolysis and OXPHOS conditions across datasets (Appendix A). Since these drugs were nominated using their imputed values in patient tumors, we went back to the initial CCL drug screen data to confirm the measure drug sensitivity was also correlated with glycolysis and OXPHOS scores. A total of 15 of the 17 drugs showed higher sensitivity among CCLs with higher glycolysis and OXPHOS features, consistent with our predictive model’s outcomes in mCRPC tumors (Appendix A). To anticipate and avoid the potential development of drug resistance, we also examined the drug-induced glycolysis and OXPHOS activities in a mCRPC cell line (PC3) using the LINCS L1000 dataset. Specifically, the glycolysis and OXPHOS scores in PC3 cells before and after each drug treatment were gauged. A total of 6 of 15 drugs did not have perturbation information; of the remaining 9 drugs, 3 would lead to higher glycolysis or OXPHOS after treatment and therefore were removed from the candidate list (Appendix A).

In summary, a total of 12 drug candidates were obtained after a four-step screening process. These drug candidates are either FDA-approved or in the late stage of drug development and were predicted to be efficacious in conditions of high glycolysis and OXPHOS. They exhibited consistently predicted activities across different clinical datasets and in measured drug response profiles of CCLs; perturbation analysis further suggested a lower risk of metabolism activation after drug treatments. Figure 2 shows the coefficient and relative *p*-value of regression analyses between glycolysis/OXPHOS scores and measured drug response AUC across CCLs for these 12 drug candidates. Compared to other drugs, P276-00 and CNF2024 showed higher significantly negative associations between glycolysis scores and measured drug responses (*p* = 0.0005, *p* = 0.0008), while ivermectin showed the most significantly negative association with OXPHOS scores (*p* = 0.0012) along with decreased glycolysis scores (*p* = 0.0093, Welch’s *t*-test) and OXPHOS scores (*p* = 0.014, Welch’s *t*-test) in PC3 cells post-treatment (Appendix A). Taken together, we prioritized P276-00, CNF2024, and ivermectin for experimental validation. These three candidate drugs were all significantly and negatively associated with glycolysis scores (Figure 3A–C and Appendix A) and OXPHOS scores (Figure 3D–F and Appendix A) in two independent clinical studies, suggesting enhanced drug efficacy under conditions of high glycolysis and OXPHOS in clinical samples.

### 2.2. Experimental Validation

To evaluate the efficacy of selected drug candidates in advanced PC with high glycolysis activity, we generated in vitro experimental models that stimulate differential glycolysis conditions (Figure 4A). To be consistent with our drug discovery pipelines and data sources, we used the CRPC cell line PC3 cells to establish glycolysis conditions. We input a significantly lower amount of glucose (1 mM of glucose) in cell culture media to mimic nutrition deprivation under the tumor microenvironment and to stimulate glycolysis activity [39], while a 5.5 mM glucose concentration was used to mimic normal physiological glucose level [40]. After 72 h, glycolytic flux (*p* < 0.05, Student’s *t*-test) and glycolytic capacity (*p* < 0.05, Student’s *t*-test) were significantly higher in PC3 cells cultured in low-glucose media compared to those cultured in normal glucose media (Figure 4B,C). This confirmed that decreasing glucose concentration, akin to the concentration within the tumor microenvironment in vivo, was able to stimulate glycolysis in PC3.

To evaluate drug efficacy, we exposed control and glycolytic PC3 cells to the three drugs of interest independently (Figure 4D). Our findings validated the predicted higher sensitivity of all three drugs in mCRPC with high glycolysis activity. Specifically, the dose–response curve of ivermectin in glycolytic PC3 cells was significantly different compared to that observed under normal conditions (*p* < 0.0001, two-way ANOVA) (Figure 4E). Similarly, we observed the same trend for CNF2024 between glycolysis and control groups (*p* < 0.0001, two-way ANOVA) (Figure 4F) as well as P276-00 (*p* < 0.0001, two-way ANOVA) (Figure 4G). No difference in the dose–response curve was observed when cells were treated with docetaxel (a negative control) (*p* = 0.0544, two-way ANOVA) (Appendix A). In addition, the half-maximal inhibitory concentration (IC_50_) of ivermectin in glycolytic PC3 cells was lower compared to its value under normal conditions (4.2 µM versus 6.4 µM), and a similar lower IC_50_ was observed for CNF2024 (40.3 nM versus 85.4 nM). In the case of P276-00, a relatively smaller difference was observed in the IC_50_ (554.5 nM versus 636.1 nM). These findings demonstrated the enhanced efficacy of the identified drug candidates when PC3 cells were subjected to a high-glycolysis condition, as opposed to a normal glycolysis status, further validating our computational drug nomination.

### 2.3. Candidate Drug Biomarker Discovery

To further facilitate the potential development of identified drug candidates, we conducted biomarker discovery, informing patient groups that may benefit from these therapeutics. Here, we presented two ways to identify patient populations for our experimental validated candidate drugs. One method was based on the phenotypic features of patients; the other was based on the expression abundance of specific genes.

#### mCRPC Patients with Low AR Expression and High Glycolysis Were Predicted to Be More Sensitive to Ivemectin and P276-00

AR low/negative subtypes of CRPC often display reduced sensitivity to androgen-deprivation therapy (ADT) [41] or ARSI therapy [42]. In addition, a previous study suggested an interplay between AR and glycolysis where patients displaying AR independence and high glycolysis had a higher risk of mortality [12]. To this end, our identified drugs may offer new therapeutic opportunities for these patients. Therefore, we examined whether a patient subgroup with such phenotypic features would be more responsive to our experimentally validated drugs. We categorized patients into two groups: AR(low)Gly(high) and all others. The distribution of patients based on their AR and glycolysis activity is displayed in Appendix A. We then conducted Welch’s *t*-test to investigate differences in predicted sensitivity to ivermectin between the two strata (Appendix A). Compared to all others, AR(low)Gly(high) patients were predicted to be more sensitive to ivermectin in both SU2C/EC (*p* = 0.032) and SU2C/WC (*p* = 0.002) clinical studies (Appendix A). Similarly, P276-00 was predicted to be more efficacious for AR(low)Gly(high) patients as well, with statistical significance in both SU2C/EC (*p* = 0.018) and SU2C/WC (*p* = 0.018) studies (Appendix A). However, we did not observe significant differences in predicted drug responses to CNF2024 between the two groups (Appendix A). These findings suggested a potential therapeutic advantage for ivermectin and P276-00 in the specific AR(low)Gly(high) patient population.

While stratification based on AR and glycolysis features associated with the predicted patient response to ivermectin and P276-00, to further identify key molecular components, we also set out to identify drug-specific marker genes for each of the drug candidates, namely, ivermectin and CNF2024, by utilizing a PPI network and topological methods. We identified a small collection of hub genes as drug-specific biomarkers based on centrality. Centrality indicates the likelihood of a gene being functionally capable of holding communicating nodes together in a biological network. Due to the pivotal role in maintaining the structural and functional integrity of the network, the expression change of hub genes likely affected therapeutic responses. For ivermectin, we observed that hub genes identified from 12 independent topological methods were highly overlapped, as shown in Figure 5A. We further refined our selection of hub genes to the top four-tier consensus genes, choosing only those identified by at least 6 out of 12 topological methods. Figure 5B presents the correlation coefficients between the expression of hub genes and the measured response to ivermectin. Among them, *RPLP0*, *RPS18*, *NPM1*, *RPS2*, *RPS27A*, and *EEF1B2* exhibited significantly negative correlation coefficients between gene expression and drug responses in CCLs, suggesting a higher predicted efficacy of ivermectin in CCLs with higher expression of these genes. In addition, these biomarker genes also showed significant correlations with OXPHOS scores in patients from the SU2C/EC clinical study (Appendix A), supporting connections between marker genes, OXPHOS scores, and the response to ivermectin. We found that patients with a high expression of *EEF1B2* exhibited a worse survival rate compared to the low-*EEF1B2* expression group (*p* = 0.033, log-rank test) (Figure 5C). This implied that mCRPC patients with a worse outcome stratified by *EEF1B2* expression abundance may benefit from ivermectin treatment.

For CNF2024, we identified a number of highly overlapped hub genes across 12 topological methods (Figure 6A). We further narrowed down the selection of hub genes to the top four-tier consensus genes, which were identified in at least 8 out of 12 topological methods. Figure 6B presents correlation coefficients between the expression of hub genes and the measured response to CNF2024 in CCLs. Among them, *CCNB1*, *RPS2*, *CCNA2*, and *CDT1* exhibited significantly negative correlation coefficients between gene expression and drug responses in CCLs, suggesting higher predicted efficacy of CNF2024 for CCLs with higher gene expression. Additionally, the gene expression of CCNB1, CCNA2, and CDT1 was corrected with glycolysis scores in patients from the SU2C/EC clinical study, implying their expression is informative of the drug response (Appendix A). Notably, when stratified by the median *CCNA2* expression, mCRPC patient tumors with a high expression of *CCNA2* exhibited worse survival compared to those with a low expression (*p* = 0.00081, log-rank test) (Figure 6C). Given its correlation with drug response, patients with high *CCNA2* and a worse clinical outcome may benefit from CNF2024. Overall, this biomarker identification approach not only proposed drug-specific marker genes but also indicated patient groups that may benefit from our drug candidates.

## 3. Discussion

In this study, we integrated computational and experimental approaches to quickly identify efficacious drugs for mCRPC patients with elevated glycolytic activity within the tumor microenvironment. We computationally predicted sensitivities to over one thousand drugs in each of the mCRPC patient tumors from two independent clinical cohorts. Our group was among the first to develop these computational approaches to construct relationships between molecular features and measured drug sensitivity data using preclinical drug screening datasets and apply them to predict patient responses to multiple drugs [31]. Since the publication of our work, many new approaches have sprung out [43,44]. These approaches were built upon many different principles and employed different strategies (some deep learning). Although small incremental improvements in prediction accuracy can be found for specific drug(s)/dataset(s), overall, there is no clear winner among all presented methods. Therefore, we chose to employ a relatively simple but well-used method (OncoPredict) for our drug nomination pipeline.

By integrating tumors’ glycolysis and OXPHOS scores with the predicted drug response, we identified efficacious candidate drugs targeting high-glycolysis conditions while considering the risk of resistance caused by metabolism rewiring to high OXPHOS. Specifically, 77 drug candidates (58 Drug_HG_ and 19 Drug_DE_) were identified in both clinical cohorts. We observed a wide spectrum of MOAs among these candidate drugs. Some of them, such as aurora kinase inhibitors, CDK inhibitors, and HSP inhibitors, have been shown to be mechanistically associated with cancer glycolysis [45,46]. Furthermore, we narrowed down our candidate drug list by first assessing development status and screening for drugs, which showed consistent results across various datasets to ensure the robustness of our findings. Meanwhile, we examined connections between measured drug response and glycolysis/OXPHOS among CCLs. Finally, we assessed the impact of drug treatment on glycolysis and OXPHOS scores using perturbation-driven CCL transcriptomic profiles. This step was used to ensure that therapeutic interventions do not inadvertently enhance metabolic pathways that could exacerbate the disease or lead to treatment resistance. Finally, based on the effect size between the measured drug responses and glycolysis/OXPHOS scores in CCLs, three drugs were prioritized to be evaluated in mCRPC cells experimentally. All three drugs showed higher efficacy in the high-glycolysis conditions in preclinical testing, echoing our computational projections. Lastly, biomarker identification was carried out. We found that ivermectin and P276-00 were predicted to be more responsive in the AR(low)Gly(high) mCRPC patient population. Furthermore, patients with a high expression of marker genes *EEF1B2* and *CCNA2* expression displayed worse survival outcomes and may benefit from ivermectin and CNF2024 treatment.

Ivermectin has been reported to achieve its anti-cancer effect through multiple diverse paths [47], including the inhibition of multidrug resistance proteins [48], the Akt/mTOR pathway [49], and other tumor progression pathways [50,51]. Relevant to this work, ivermectin has been shown to achieve anti-cancer effects by impacting cancer energy metabolism in several malignancies. In glioma, ivermectin inhibits glycolysis by decreasing the expression of glucose transporter 4 (GLUT4), one of the key transporters in glycolysis [52]. In renal cancer, ivermectin decreases the mitochondrial membrane potential, mitochondrial respiration, and ATP generation, leading to mitochondrial dysfunction and oxidative damage [53]. A similar effect was also observed in breast cancer cells [54]. All of these results are consistent with our findings in mCRPC and support the use of ivermectin to combat high-glycolysis mCRPC.

CNF2024 is an HSP90 inhibitor, a class of drugs that is known to modulate the stability of androgen receptor (AR) [55]. A study co-administrating the HSP90 inhibitor 17-allylamino-17-demethoxygeldanamycin (17-AAG) and enzalutamide has demonstrated enhanced inhibition of PC tumors by disrupting AR protein stability, thereby presenting a promising therapeutic intervention for mCRPC [56]. Despite in vivo tumor inhibition results observed with 17-AAG, clinical trials have revealed challenges related to poor bioavailability and significant toxicity [57]. Consequently, there has been a concerted effort to explore alternative small-molecule HSP90 inhibitors, among which CNF2024 has shown considerable promise [58]. In Hodgkin’s lymphoma, CNF2024 has exhibited efficacy, resulting in a statistically significant decrease in tumor size [58]. Furthermore, in vitro cell viability studies conducted on L540 and L540cy cell lines showed that CNF2024 had IC_50_ values more than 7.5 times lower than that of 17-AAG, underscoring its potential as a more effective therapeutic agent [58]. Another HSP90 inhibitor, SU086, has been identified for its direct impact on glycolysis and its inhibitory effects on PC tumor invasion, migration, and growth [38]. Combining SU086 with enzalutamide and abiraterone in CRPC cell lines has revealed synergistic and additive effects, respectively [38]. In our pipeline, we initially identified two HSP 90 inhibitors (ganetespib and CNF2024) as drug candidates in both mCRPC patient cohorts. As expected, both ganetespib and CNF2024 exhibited a negative correlation between glycolysis scores and measured sensitivity toward these drugs in CCLs. However, ganetespib sensitivity was positively correlated with OXPHOS scores in CCLs, suggesting a weakened efficacy of ganetespib in a high-OXPHOS condition. Interestingly, docetaxel, one of the standard care therapies for mCRPC, has been shown to induce metabolism reprogramming from glycolysis to OXPHOS in PC [59]. A previous clinical trial of ganetespib in docetaxel-pretreated mCRPC patients showed a limited clinical benefit [60], implying a potential drug insensitivity, likely in part, from high OXPHOS. Unlike ganetespib, our findings suggest CNF2024 may retain its efficacy even under metabolic reprogramming to high OXPHOS, further warranting clinical evaluation of this drug.

ARSIs, such as enzalutamide and abiraterone, have significantly improved clinical treatment outcomes of CRPC. Yet, intense treatments of ARSIs often result in therapy resistance partially due to a reduction in AR signaling [61] or AR expression [62]. About 15–20% of CRPC patients treated with ARSIs eventually transdifferentiate into a more lethal disease, neuroendocrine prostate cancer (NEPC) [63,64]. Previous studies have suggested that metabolism reprogramming drives NEPC differentiation [65] and have identified elevated glycolysis [66] and low AR signaling activity as biological features of NEPC [61]. In our biomarker discovery based on the AR and glycolysis activities, we identified ivermectin and P276-00 as possessing higher efficacy in the AR(low)Gly(high) mCRPC population. This warranted a follow-up study of these drugs to delay the disease progression of ARSI-pretreated CRPC to NEPC.

Many ivermectin-associated hub genes identified through our biomarker discovery pipeline are ribosome synthesis-related genes, such as *RPLP0*, *RPS18*, *RPS2*, *RPS27A*, and *EEF1B2*. Studies have shown ribosome synthesis is relevant to tumor progression and therapeutic resistance [67]. Here, several ivermectin biomarkers identified in our work have been reported to play a critical role in PC. For instance, *RPLP0* has been identified as a key regulator in PC [68], while *RPS18* is linked to PC recurrence and prognosis [69]. *RPS2* has been recognized as a highly selective therapeutic target for PC; knocking down *RPS2* expression using oligonucleotides in an in vivo PC3-3ML model led to a promising PC eradication [70]. Beyond PC, one of the identified hub genes, *NPM1*, has significant associations with glycolysis in lung cancer [71,72] and pancreatic cancer [73]. Among these ribosome synthesis-related genes, we observed a correlation between these hub genes and OXPHOS. Studies have found that OXPHOS caused MCF7 cells to be more sensitive to ivermectin [54], and ivermectin exhibited a promising tumor growth inhibition through inducing mitochondrial dysfunction and oxidative stress in renal cell carcinoma highly reliant on OXPHOS for survival [53]. This evidence supported our identified predicted drug response biomarkers for ivermectin. Taken together, hub genes identified by our study are connected with ivermectin’s mechanism of action and play a crucial role in the context of PC and cancer energy metabolism.

In biomarker discovery for CNF2024, cell cycle-related genes such as *CCNB1*, *CCNA2*, and *CDT1* were identified. Previous studies have shown that CNF2024 induced G_2_ cell cycle arrest in Hodgkin’s lymphoma cells [58] and decreased cell cycle-related proteins, including CDK1, CDK2, and cyclin D3, along with either G_1_ or G_2_ cell cycle arrest in multiple lymphoma cell lines [74]. The relationship between HSP90 and the cell cycle is highly connected since HSP90 directly regulates key cell cycle regulator proteins [75]. Beyond the cell cycle, we found that the identified biomarkers are relevant to glycolysis. For example, *CCNB1* has been identified as one of the top genes strongly correlated with glycolysis in various cancers [76]. Additionally, Jiang et al. stratified patients with hepatocellular carcinoma (HCC) into low and high risk based on a glycolysis signature and found the cell cycle to be one of the enriched pathways from differentially expressed genes among high-risk patients [77]. Furthermore, under hypoxia, a factor that induces glycolysis, *CCNA2* expression was upregulated in acute myeloid leukemia [78], and a similar relationship between hypoxia and *CDT1* was found in lung cancer as well [79]. In addition, cyclin A (encoded by *CCNA2*) expression was suppressed when the platelet isoform of phosphofructokinase 1 (PFKP), a key glycolysis enzyme, was knocked down in colorectal cancer cells [80]. Notably, *CCNA2* has been identified as a prognostic factor in patients with PC [81]. Overall, these connections between glycolysis and hub genes align with our research findings.

Biomarkers identified in the current study were derived independently from multiple computational methodologies. Yet, they have not been experimentally validated. This warrants subsequent studies to functionally study the role of genes and the drug of interest.

## 4. Materials and Methods

### 4.1. Data Acquisition and Preprocessing

The Cancer Therapeutics Response Portal Version 2 (CTRPv2) [82] drug response database was obtained from the Cancer Target Discovery and Development (CTD2) Network’s data portal, maintained by the National Cancer Institute’s Office of Cancer Genomics. The database is accessible via the following website: https://ocg.cancer.gov/programs/ctd2/data-portal (accessed on 19 December 2021). The normalized Profiling Relative Inhibition Simultaneously in Mixtures (PRISM) area under the dose–response curve (AUC) data was downloaded from https://oncotherapyinformatics.org/simplicity/ (accessed on 3 August 2021) [83]. CCLE gene expression data were sourced from the Broad Institute’s Cancer Cell Line Encyclopedia (CCLE) through the Dependency Map (DepMap) portal [84]. The gene expression profile of mCRPC tumors from the Standard Up to Cancer East Coast (SU2C/EC) study was accessed through the cBioportal [85]. Originally, the unit of gene expression data for the SU2C/EC patients was fragments per kilobase per million (FPKM) and was preprocessed as log2(FPKM + 1). Additionally, we obtained gene expression data of mCRPC tumors from an independent clinical study, namely, the Standard Up to Cancer West Coast (SU2C/WC) [86]. The unit of gene expression data for mCRPC tumors from the SU2C/WC study was originally transcript per million (TPM) and was preprocessed as log2(TPM + 1). Perturbation-driven gene expression profiles of PC3 cells were downloaded from the Connectivity Map (CMAP) [87]. All computational analyses were performed with R, version 4.1.0. The R scripts, along with the relevant data, can be accessed at https://osf.io/nt4vu/.

### 4.2. Impute Drug Response in Patients with mCRPC

To impute drug responses in patients with mCRPC, we employed a computational method OncoPredict [30,31], which leverages large-scale drug screens on CCLs to estimate sensitivities to various drugs in patient tumor datasets through ridge regression. In this study, we imputed drug responses for approximately 1900 agents, from CTRPv2 and PRISM, in each mCRPC patient tumor from two different clinical cohorts (namely, the SU2C/EC and the SU2C/WC). The following data preprocessing parameters were defined for employing OncoPredict. First, common genes between mCRPC patients and CCLs were selected and homogenized by ComBat [88] to remove batch effects. Second, power transformation was applied to CCLs’ measured drug response data, and then feature selection was performed to filter for the top 50% of the genes with the highest variation across samples. After model training from CCLE gene expression (predictors) and drug responses (dependent variables), the estimated coefficients of the trained model were used to predict the likelihood of patients’ drug responses. A lower value of predicted drug response (AUC) suggests increased sensitivity to the specific drug.

### 4.3. Regression Analysis to Identify Efficacious Drugs for mCRPC with High Glycolysis

The objective of this research study is to identify drugs efficacious for mCRPC patients characterized by elevated glycolysis. To achieve that goal, data analysis was performed to find two types of drugs: (1) drugs efficacious for high glycolysis (drug_HG_) and (2) drugs efficacious for both high glycolysis and OXPHOS, called dual-effect drugs (drug_DE_). The former one is intended to inhibit high-glycolysis tumor growth, and the latter one is to inhibit both high glycolysis and OXPHOS tumor growth to minimize the risk of drug resistance development from metabolism rewiring to OXPHOS. We first calculated glycolysis scores of mCRPC tumor samples from both SU2C/EC and SU2C/WC studies by using the R package GSVA [89] and then constructed linear regression models to probe the relationship between patients’ glycolysis/OXPHOS scores and their predicted drug response, as described in Equations (1) and (2). Of note, since the age of procurement is significantly associated with OXPHOS scores in the SU2C/EC study (*p* = 0.0252, F-test), we adjusted for the age of procurement in the OXPHOS linear regression model if information on the age of procurement was given.
(1)$$Glycolysis\;Score=\beta_{0}+\beta_{1}\times Predicted\;Drug\;Response$$
(2)$$\begin{array}{ll} OXPHOS\;Score & =\beta_{0}+\beta_{1}\times Predicted\;Drug\;Response\\ & +\beta_{2}\times Age\;at\;Procurement \end{array}$$

Here, GlycolysisScore represents a GSVA glycolysis signature score of each mCRPC tumor based on 40 glycolysis-related genes [90] on a scale of 0–100; OXPHOSScore represents a GSVA OXPHOS signature score of each mCRPC tumor based on the hallmark OXPHOS gene set [91] on a scale of 0–100; PredictedDrugResponse indicates the predicted sensitivity of a patient to a specific drug; AgeatProcurement is the patient’s age when a tumor sample was taken from them. $\beta_{0}$ is the intercept of a linear regression model; $\beta_{1}$ is a coefficient of predicted drug response of mCRPC tumors; $\beta_{2}$ is a coefficient of the age of procurement.

Each drug has its linear regression model. Only the drugs that show a significantly negative coefficient $\beta_{1}$ with a false discovery rate (FDR) of less than 0.05 in both SU2C/EC and SU2C/WC clinical studies were included for subsequent analysis (Appendix A). Herein, a drug with a negative coefficient $\beta_{1}$ implies that high-glycolysis/OXPHOS mCRPC tumors are more sensitive to the drug compared to low-glycolysis/OXPHOS mCRPC tumors.

### 4.4. Selecting Drug Candidates with Higher Robustness for Validation

To narrow down our selection of drug candidates in two clinical studies, we implemented four stringent filtering criteria to ensure the selection of robust and clinically relevant drugs. First, priority was given to drug candidates that have received FDA approval or have been advanced into clinical trials, emphasizing had regulatory status and the potential for rapid clinical application. Second, we eliminated drugs that exhibited significantly opposing correlations between their predicted drug responses and glycolysis or OXPHOS scores when analyzed across different datasets. Third, we focused on drugs that consistently demonstrated a negative correlation between the measured drug response AUCs and both glycolysis and OXPHOS scores in pan-cancer CCLs. Lastly, we minimized the risk of metabolic activation post-treatment by excluding drugs that significantly increase glycolysis or OXPHOS scores after 24 h of treatment in human prostate cancer cell line PC3 based on perturbation-driven gene expression profiles sourced from the Connectivity Map (CMAP) [87].

### 4.5. Cell Culture and Reagents

PC3 cells (ATCC) were maintained in F-12K (ATCC) media with 10% FBS (Thermo Fisher Scientific, Gibco, Waltham, MA, USA) at 37 °C with 5% CO_2_. PC3 models of high and normal glycolysis were achieved by culturing PC3 cells in two different glucose concentration media (1 mM and 5.5 mM). Glucose conditional medium was made using DMEM (Thermo Fisher Scientific, Waltham, MA, USA, Cat. No. A1443001) supplemented with 1 mM or 5.5 mM of glucose, respectively, with 2 mM of glutamine and 10% FBS. Ivermectin (CAS No. HY-15310), CNF2024 (CAS No. HY-10212), P276-00 (CAS No. HY-16559), and docetaxel (CAS No. HY-B0011) for drug candidate experimental validation were sourced from MedChem Express (Monmouth Junction, NJ, USA) and dissolved in dimethyl sulfoxide (DMSO) to achieve 5 mM drug stock solutions. The kit for the Glycolysis Stress Test was obtained from Seahorse Bioscience (Billerica, MA, USA).

### 4.6. Cell Growth and Viability

PC3 cells were harvested and stained with Hoechst prior to seeding at a density of 8000 cells per well into 96-well tissue culture plates. Cells were allowed to attach for 24 h while incubating at 37 °C with 5% CO_2_ prior to changing the culture medium to either a low-glucose (1 mM) or a normal-glucose (5.5 mM) conditional medium. Culture plates with the two conditional media were incubated for 24 h prior to the addition of candidate compounds at concentrations ranging from 2.3 µM to 10 µM for ivermectin, 1.6 nM to 400 nM for CNF2024, 0.2 µM to 6 µM for P276-00, and 10 nM to 500 nM for docetaxel (as a negative control). Over the duration of the drug exposure, 1 mM of glucose was added every 24 h to maintain the same level of glucose difference between the two conditional media and prevent cell death under nutrition stress conditions. Cell counts were obtained longitudinally every 24 h through the 96 h drug exposure utilizing the Cytation I Live Cell Imaging System (Agilent BioTek, Santa Clara, CA, USA). To account for cell seeding differences, prior to the addition of the candidate drug, cell counts were transformed by subtracting the baseline cell count obtained 24 h after the addition of the conditional glucose media (delta cell count). Delta cell counts for each drug exposure condition at each time point were normalized to the no-drug control condition to obtain the percentage of normalized cell count. A two-way ANOVA was conducted to analyze the significance of the percentage of normalized cell count differences at 48 h between low- and normal-glucose groups.

### 4.7. Seahorse XFp Glycolysis Stress Test Assay

To measure the glycolysis activity of PC3 cells in low- and normal-glucose conditional media, we used a Seahorse XF cell glycolysis stress test kit and conducted the test using Agilent Seahorse XFe96 analyzer (Seahorse Bioscience, Agilent Technologies, Inc., North Billerica, MA, USA) according to the manufacturer’s protocol. After cells were pre-cultured in conditional media for 48 h where 1 mM of glucose was added per 24 h to prevent cells from dying under the nutrition stress condition, 1.4 × 10^4^ cells were plated directly onto Seahorse XFp plates to a final volume of 80 µL overnight to reach total 72 h in conditional medium before measuring glycolysis activity. Until cells reached 72 h of culturing in a conditional medium, the medium was replaced with a 2 mM of glutamine-supplemented Seahorse base medium, pH 7.4, and incubated in a deoxygenated 37 °C incubator for one hour. Additionally, flux cartridges and wells were prepared by hydrating them overnight in a deoxygenated incubator before conducting the assays. Immediately before conducting the Seahorse glycolysis stress test, the medium was replaced with a final volume of 180 µL of the fresh Seahorse medium according to the manufacturer’s protocol. To prepare the Seahorse glycolysis stress test, 20 μL of glucose (10 mM), 22 μL of oligomycin (1 µM), and 25 μL of 2-DG (50 mM) were first loaded into ports A, B, and C of the flux cartridge according to the manufacturer’s recommendations. The first hour in the glucose-free XF Seahorse medium is for measuring the basal level of medium acidification. After that, glucose was injected to measure glycolysis medium acidification, followed by oligomycin to measure maximum glycolytic capacity and 2-DG to calculate glycolytic reserve sequentially. In the end, cells were stained with Hoechst 33342 (118 μg/mL final concentration), and the fluorescence field was scanned by a Cytation 1—Cell Imaging Multimode Reader (Agilent BioTek, Santa Clara, CA, USA) and normalized based on cell counts. This glycolysis-relevant metabolism activity was described by using the extracellular acidification rate (ECAR). Glycolytic flux is the ECAR value of glucose-treated cells minus the baseline (time point 6 minus time point 3). Glycolytic capacity was determined by subtracting the baseline ECAR value at time point 3 from the ECAR value at time point 9 after treatment with oligomycin. A Student’s *t*-test was used to analyze the significance of the glycolysis flux and the glycolysis capacity differences between low- and normal-glucose groups.

### 4.8. Stratifying Patients Based on AR and Glycolysis Status for Biomarker Discovery

We stratified patients into two distinct groups: those with low AR activity and high glycolysis (AR(low)Gly(high)) and all others. Subsequently, we compared the predicted drug responses of the experimentally validated candidate drugs (ivermectin, CNF2024, and P276-00) between these two groups.

To determine AR activity, for each patient, an AR activity score was calculated using a set of signature genes [92]. The log2-FPKM values (for the SU2C/EC study) or log2-TPM values (for the SU2C/WC study) of each signature gene were normalized across all tumor samples to z-scores. A patient’s AR activity score was then derived by summing up all z-scores. Scores across patients were scaled to range between 0 and 1, with a higher value indicating higher AR pathway activity. AR(low) was characterized by an AR score lower than the median of AR scores across all patients, while Gly(high) was defined as a glycolysis score higher than the median of glycolysis scores. A Student’s *t*-test was used to analyze the significance of predicted drug response differences between AR(low)Gly(high) and other groups.

### 4.9. Biomarker Discovery through PPI Network

To identify the potential biomarkers for validated drug candidates, we initially identified genes that exhibited a strong correlation with predicted drug responses using a correlation coefficient of 0.4 as a threshold to ensure a focus on the most relevant genetic interactions. Afterward, a protein–protein network (PPI) comprising these highly correlated genes was constructed using the Search Tool for the Retrieval of Interacting Genes/Proteins (STRING v12: https://string-db.org/ (accessed on 20 September 2023)) [93], with a confidence score of ≥0.7 and a maximum number of interactors set to zero. The constructed PPI network was subsequently analyzed using Cytoscape software (version 3.9.1) [94] with the cytoHubba plugin [95] to identify the top 50 hub genes from 12 topological analysis methods independently. The 12 topological analysis methods include Betweenness, Bottleneck, Closeness, Clustering Coefficient, Degree, Density of Maximum Neighborhood Component, EcCentricity, Edge Percolated Component, Maximal Clique Centrality, Maximum Neighborhood Component, Radiality, and Stress. The resultant 50 hub genes from independent topological methods were then overlapped to derive consensual hub genes. To further prioritize the consensual hub genes relevant to drug sensitivity, Spearman’s rank correlation analysis between gene expression level and measured drug response AUC in CCLs was conducted. Genes exhibiting a significantly negative Spearman correlation (*p* < 0.05) were chosen as potential biomarkers for drug candidates. This selection indicates that higher expression levels of these genes correlate with higher drug efficacy in CCLs.

### 4.10. Survival Analysis

Since biomarker discovery is identified based on the level of connectedness of highly correlated genes in the PPI network, there is a high chance of the identified biomarkers possessing an important functional feature, such as relevance to disease progression. To investigate that, patients with mCRPC from the SU2C/EC clinical study were categorized into two groups, a low-expression group and a high-expression group based on the median of gene expression levels. Survival analysis was then conducted, comparing these two patient subpopulations. Survival probabilities were calculated by the Kaplan–Meier method, and group comparisons were performed using the log-rank test.

## 5. Conclusions

In summary, we efficiently identified and validated multiple drugs, each with different MOAs, demonstrating efficacy in treating mCRPC characterized by heightened glycolysis in the tumor microenvironment. Our biomarker study also revealed several potential biomarkers for patient stratification when using these newly nominated drugs. The significance of this study is to expand new efficacious therapeutics beyond traditional androgen deprivation therapies by precisely targeting the mCRPC patient population with a unique biological feature, high glycolysis. These newly nominated drugs and biomarkers, once clinically validated, can improve the treatment outcomes of patients with mCRPC. Looking ahead, the insights gained here could inform the development of more targeted, effective, and personalized treatment strategies, revolutionizing the care for patients with this aggressive form of PC.

## Figures and Tables

**Figure 1 pharmaceuticals-17-00569-f001:**
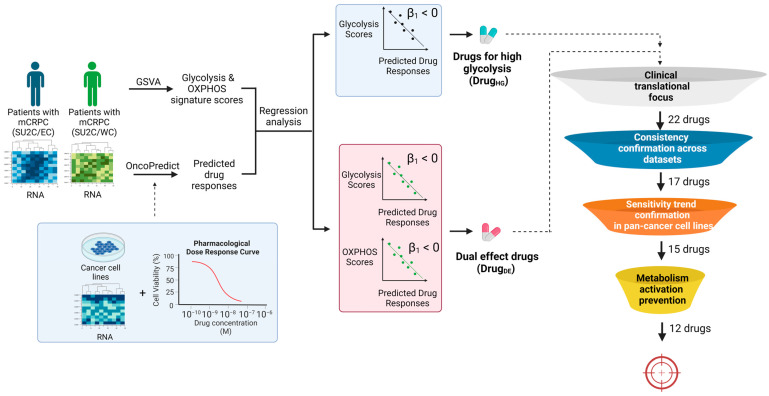
Lead compound nomination pipeline for mCRPC patients with high glycolysis activity. OncoPredict was used to impute sensitivities of mCRPC patient tumors to various drugs. GSVA was used to compute glycolysis and OXPHOS scores for each mCRPC patient tumor. Subsequently, regression analysis was conducted. Drugs with predicted drug response significantly and negatively correlated with glycolysis scores or both glycolysis and OXPHOS scores were selected as drugs for high glycolysis (Drug_HG_) and dual-effect drugs (Drug_DE_), respectively. Only drugs identified in both SU2C/EC and SU2C/WC studies were included for subsequent filtering steps to select final drug candidates. The four filtering steps included (1) clinical translational focus, (2) consistency confirmation across datasets, (3) sensitivity trend confirmation in pan-cancer cancer cell lines, and (4) metabolism activation prevention. Abbreviations: mCRPC: metastatic castration-resistant prostate cancer; SU2C/EC: Standard Up to Cancer East Coast; SU2C/WC: Standard Up to Cancer West Coast; GSVA: Gene Set Variation Analysis; β_1_: a coefficient of predicted drug response of mCRPC tumors; OXPHOS: oxidative phosphorylation.

**Figure 2 pharmaceuticals-17-00569-f002:**
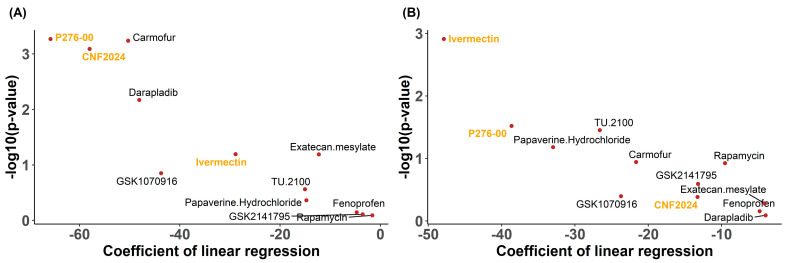
Dot plots of coefficients for the final drug candidates in pan-cancer CCLs. (**A**) Coefficients of the linear regression between glycolysis score and measured drug response AUC. (**B**) Coefficients of the linear regression between OXPHOS score and measured drug response AUC. The statistical analysis was performed using Student’s *t*-test. Drug candidates for experimental validation are labeled orange.

**Figure 3 pharmaceuticals-17-00569-f003:**
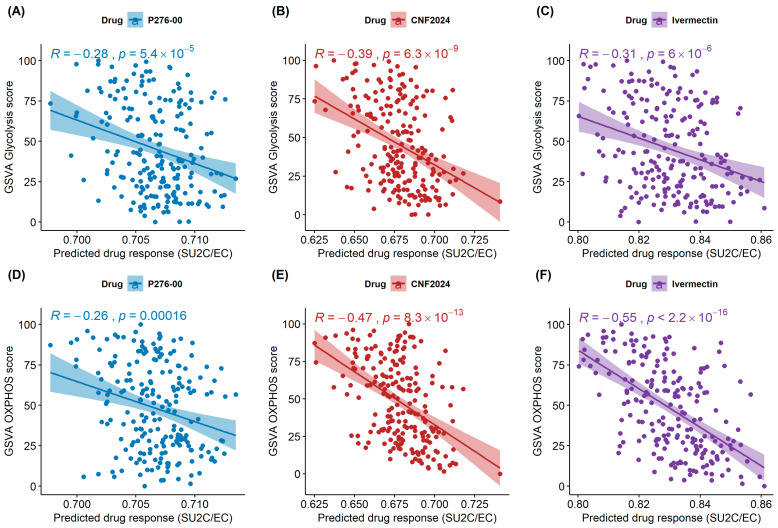
The correlation between glycolysis scores or OXPHOS scores with the predicted drug response in mCRPC patients in SU2C/EC clinical studies. (**A**–**C**) Correlations between the glycolysis score and the predicted patient response to P276-00, CNF2024, and ivermectin. (**D**–**F**) Correlations between the OXPHOS score and the predicted patient response to P276-00, CNF2024, and ivermectin. For each correlation plot, the correlation coefficient and its *p*-value are given. The statistical analysis was performed by Spearman rank correlation and Student’s *t*-test with a significance level of α = 0.05. Abbreviations: OXPHOS: oxidative phosphorylation.

**Figure 4 pharmaceuticals-17-00569-f004:**
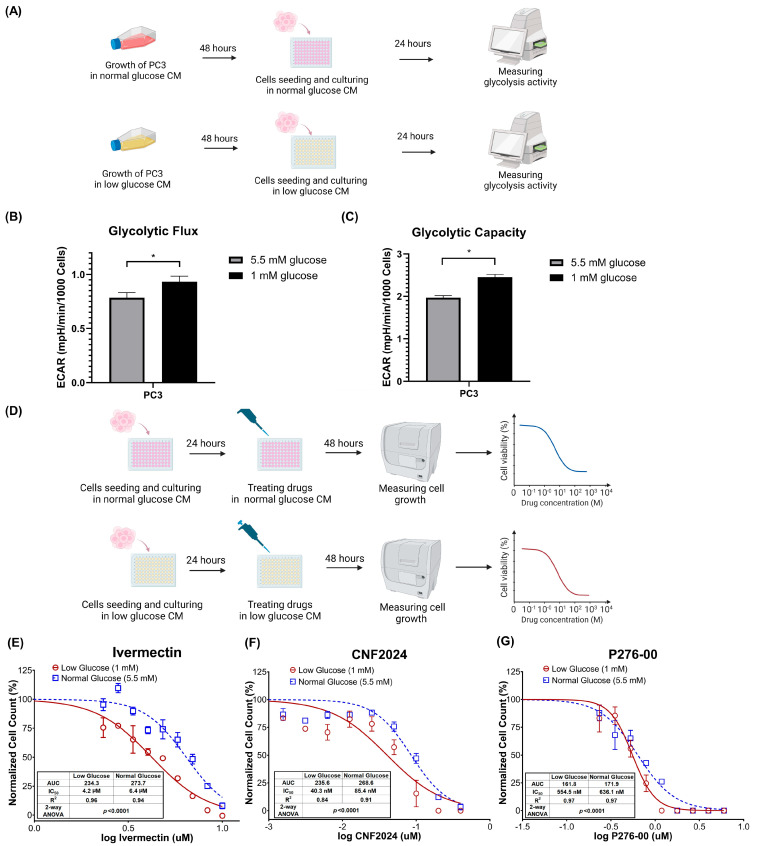
Experimental validation of candidate drugs in PC3 models with normal- and high-glycolysis conditions. (**A**) Workflow of development of glycolytic PC3 cells and stress test assays. (**B**) Glycolytic flux and (**C**) glycolytic capacity of PC3 after culturing in conditional media (CM) for 72 h. * *p* < 0.05, Student’s *t*-test. Representative results of three independent experiments with 8 to 10 replicates per group. (**D**) Workflow of drug sensitivity evaluation. (**E**–**G**) Dose–response curve of PC3 after ivermectin, CNF2024, and P276-00 treatment for 48 h, respectively, in either normal-glucose or low-glucose media. There were 3 replicates per group, and the experiments were repeated 3 times. The statistical analysis was performed using two-way ANOVA. Abbreviations: CM: conditional media; ECAR: extracellular acidification rate.

**Figure 5 pharmaceuticals-17-00569-f005:**
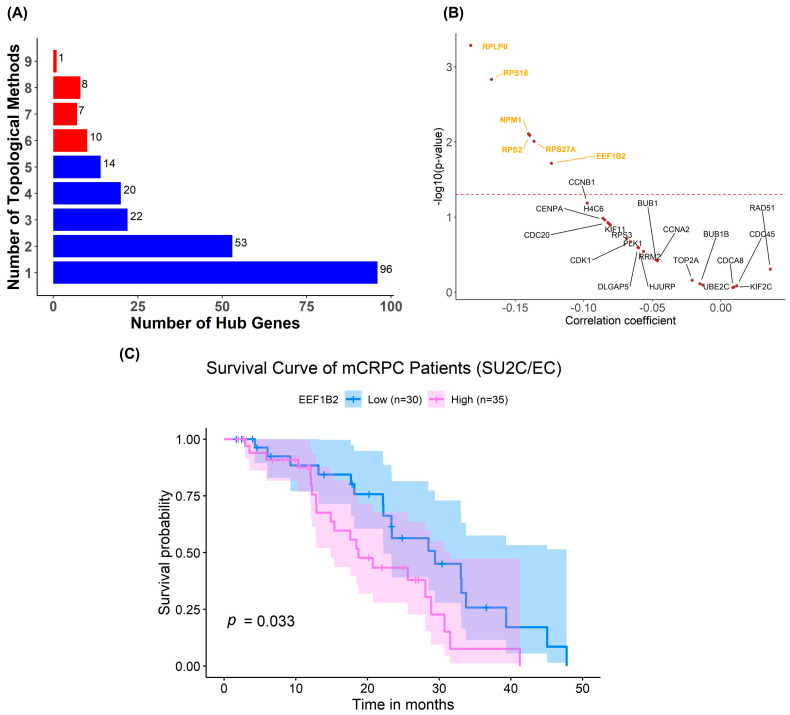
Biomarker discovery for ivermectin. (**A**) Bar plot about a number of consensual hub genes identified across 12 topological analyses. The red bars indicate the top four-tier consensus genes. (**B**) Spearman correlation analysis between gene expression and measured area under the drug-response curve (AUC) in pan-cancer CCL. *p* = 0.05 (dashed red line). The statistical analysis was performed using Spearman rank correlation and Student’s *t*-test with a significance level of α = 0.05. (**C**) Survival analysis of patients stratified based on *EEF1B2* expression level in the SU2C/EC clinical cohort. The statistical analysis was performed using the log-rank test with a significance level of α = 0.05. Abbreviations: SU2C/EC: Standard Up to Cancer East Coast.

**Figure 6 pharmaceuticals-17-00569-f006:**
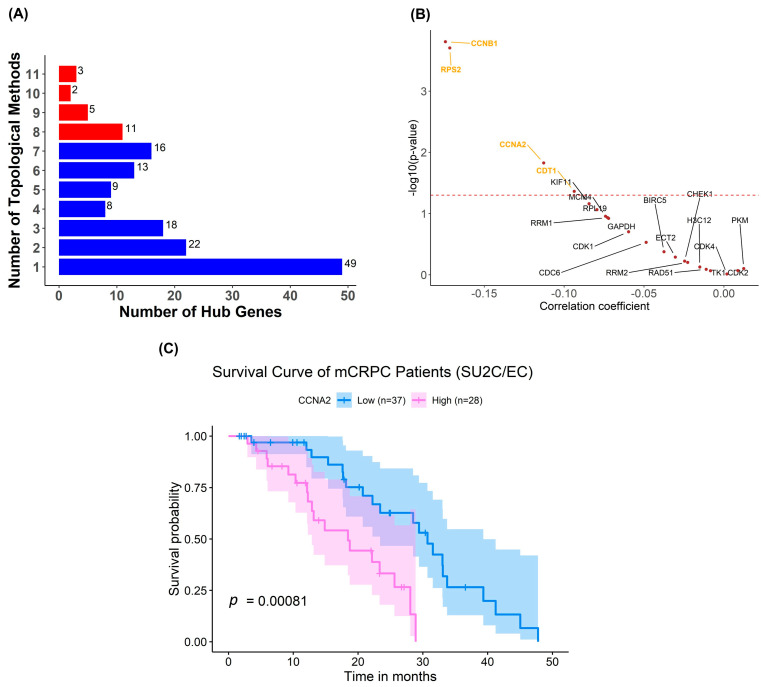
Biomarker discovery for CNF2024. (**A**) Bar plot about a number of hub genes identified across 12 topological analyses. The red bars indicate the top four-tier consensus genes. (**B**) Spearman correlation analysis between gene expression and measured area under the drug-response curve (AUC) in pan-cancer CCLs. *p* = 0.05 (dashed red line). The statistical analysis was performed by Spearman rank correlation and Student’s *t*-test with a significance level of α = 0.05. (**C**) Survival analysis on patients stratified based on *CCNA2* expression level in SU2C/EC clinical cohort. The statistical analysis was performed using the log-rank test with a significance level of α = 0.05. Abbreviations: SU2C/EC: Standard Up to Cancer East Coast.

**Table 1 pharmaceuticals-17-00569-t001:** Summary of number of drug candidates. (A) Drugs for high-glycolysis mCRPC; (B) dual-effect drugs.

**(A) Drugs Showing High Efficacy in High-Glycolysis mCRPC Patients (Drug_HG_)**				
Clinical study	SU2C/EC	SU2C/WC	SU2C/EC	SU2C/WC
Training Drug database	CTRPv2	CTRPv2	PRISM	PRISM
Drugs predicted to show high efficacy in high-glycolysis mCRPC patients	41	5	260	67
Drug_HG_ (SU2C/EC ∩ SU2C/WC)	4		54	
Drug_HG_ in primary MOA ^#,^*	0		9 ^&^	
**(B) Dual-Effect Drugs (Drug_DE_)**				
Clinical study	SU2C/EC	SU2C/WC	SU2C/EC	SU2C/WC
Training drug database	CTRPv2	CTRPv2	PRISM	PRISM
Drugs for high glycolysis mCRPC (G)	41	5	260	67
Drugs for high-OXPHOS mCRPC (O)	80	59	530	492
Drug_DE_ (G ∩ O)	13	0	156	27
Drug_DE_ (SU2C/EC ∩ SU2C/WC)	0		19	

Notes: ^#^: Primary MOA is characterized by a MOA that appeared at least twice across these 54 Drug_HG_ from PRISM. *: Primary MOAs include aurora kinase inhibitors, CDK inhibitors, EGFR inhibitors, HSP inhibitors, mTOR inhibitors, and topoisomerase inhibitors. ^&^: Drugs were obtained from overlapping the top 50% significant drugs identified from SU2C/EC and SU2C/WC studies. Abbreviations: mCRPC: metastatic castration-resistant prostate cancer; Drug_HG_: drugs for high-glycolysis mCRPC; Drug_DE_: dual-effect drugs; SU2C/EC: Standard Up to Cancer East Coast; SU2C/WC: Standard Up to Cancer West Coast; CTRPv2: Cancer Therapeutics Response Portal Version 2; PRISM: Profiling Relative Inhibition Simultaneously in Mixtures; MOA: mechanism of action; OXPHOS: oxidative phosphorylation.

## Data Availability

All data used in this study is publicly available as listed in the method section. All data and code can be found at https://osf.io/nt4vu/.

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
