# Peer review of "Computational Modeling to Identify Drugs Targeting Metastatic Castration-Resistant Prostate Cancer Characterized by Heightened Glycolysis"

_pharmaceuticals, 2024, doi:10.3390/ph17050569_

Round 1

Reviewer 1 Report

Comments and Suggestions for Authors

Authors are required to address the follwing issues:

- Include obtained results in the abstract along with a brief description of the methodology.

- Highlight additional contributions in the last paragraph of the introduction.

- Present a dataset table with patient details such as age variations, numbers, and features.

- Incorporate the complete methodology using a block diagram in the materials and methods section.

- Ensure every column in Table 1 has a specific value or use (N/A) if not present.

- Provide detailed explanations of prediction technology/algorithms.

- Compare the proposed methodology with existing approaches using a state-of-the-art table in the discussion section.

- Utilize evaluation assessment metrics for presenting results effectively.

Comments on the Quality of English Language

There are several grammatical, typographical, and spelling errors in the manuscript. Please correct them.

Reviewer 2 Report

Comments and Suggestions for Authors

Comments to the Author

In Manuscript title “Computational modeling to identify drugs targeting metastatic castration-resistant prostate cancer characterized by heightened glycolysis” computational framework has been developed to identify new therapeutic agents for Metastatic castration-resistance prostate cancer (mCRPC).  Using this framework, 77 drugs have been predicted to be more sensitive in high glycolysis mCRPC tumors. In addition, the authors also performed performed in vitro validation on three of the candidates. The manuscript is very well written, scientifically sounds and cover all methods.   This manuscript could be considered publishing after minor revision.

Comments:

[1] The author could provide a better Figures with high resolution. All figures needed better resolution. It is hard to understand the current figures for reader.

[2] In figure 1 legend some text size has different format. Please correct it. The text size should be same throughout the manuscript.

[3] What properties of drugs have been used in analyses? Whether, the authors used the Descriptors/Fingerprints or any other properties?

[4] If possible the authors should provide the applicability domain and diversity of the drugs used in this study and also discussed about the diversity. As these factors are important in the development of any machine learning frame work.

[5] Please check the plagrism as per the iThenticate report (Submitted on journal website) Percent match is 37%. Please try to reduce it.

Reviewer 3 Report

Comments and Suggestions for Authors

The goal of this manuscript is to repurpose some existing drugs/drug candidates for treating metastatic castration-resistance prostate cancer (mCRPC) by targeting tumors with high glycolysis.  The authors also tried to identify biomarkers for predicting/directing future uses of the nominated drug candidates. Therefore, the manuscript is relevant to the scope of the special issue.  The approach utilized in the research is primarily large dataset-based bioinformatics in conjunction with some in vitro experimental validation in cell viability, which is quite novel.  The paper is well organized and written.  It is easy to follow.  The claimed conclusions are largely supported by the presented results. However, the significance of the work is limited by small (<= 2-fold, Figure 4) sensitivity difference of the tested three drugs between normal and high glycolysis conditions.  The actual efficacy of these drug candidates also needs to be further validated in vivo and even in clinical settings.  Nevertheless, the paper presented some interesting initial results that warrant further investigation, which makes it suitable for publication in the journal if the authors could address the following point.

The point to be addressed:

Could the identified biomarkers be validated in vitro, such as knockdown or overexpression?  If such validation would not be helpful, could the authors address why?

Reviewer 4 Report

Comments and Suggestions for Authors

This work presents an original and interesting approach to drug repurposing for cancer treatment.  The objective is clearly stated and the introduction is well-written. However, the methodology, results, and conclusions sections require improvement, as well as some details in the discussion section. In this sense, my observations are the following:

Line 28: Change 'ivermectin' to 'Ivermectin'.

Line 40: Change 'nmol/L' to 'nmol/L'. Review the entire manuscript for correct use of units of measurement. Also, revise throughout the document for correct wording of measurement values. For example, on line 211, the correct format is '2 mM' (it is incorrect to write '2mM').

Line 136: Is the name of 'R' referring to the R package GSVA? 

Line 260: It is incorrect to use the unit 'ml'. It should be 'mL'. 

Lines 335 to 337: The font size needs to be corrected (please review the entire manuscript).

The description in Table 1 should include the meaning of all abbreviations and symbols used.

How many SU2/EC and SU2/WC patients (or cancer samples) were considered in this study? Did you consider only patients with mCRPC?

In paragraph line 358 through line 373, how do the drug data in this paragraph relate to Table 1? At the beginning of this paragraph, the authors refer to Figure 1, however, Figure 1 does not show data on number of drugs used. What is described in this paragraph is related to the supplementary figure that does not appear in the manuscript.

It should be considered the importance of including in the manuscript the data or schemes that are in the supplementary material.

In the description of Figure 2 there is no mention of what the yellow color of certain drugs means. Figure 2 should mention the name of the statistical test used.

Figure 3 should indicate the type of statistical test used and the p value with statistical significance.

Figure 4 does not mention which statistical test was used in figures B, C, E, F and G.

In section 3.3 Candidate drugs biomarker Discovery (Lines 444 to 448), their results are not presented.

In all the supplementary figures, the wording of their description should be improved, the statistical tests used and the meaning of the abbreviations included should be indicated.

Figure 6 should mention which statistical test was used and the meaning of the abbreviations included.

In the results section, the authors do not mention which proteins are encoded by the genes most expressed by the use of ivermectin (RPLP0, RPS18, NPM1, RPS2, RPS27A and EEF1B2) and CNF2024 (CCNB1, CCNA2, and CDT1).

Lines 556 and 557: GLUT4 is not considered an enzyme, nor is it a protein of glycolysis. It is a transporter. This wording should be corrected.

Lines 564 and 565: the authors should indicate what the abbreviation 17-AAG means.

Line 633: What does the abbreviation PFKP stand for? Is it phosphofructokinase 1 or phosphofructokinase 2? Phosphofructokinase 2 is not part of glycolysis.

In their conclusion the authors do not mention drugs and biomarkers with a greater potential effect in their use for the treatment of mCRPC.

A good part of the results presented by the authors are based on supplementary figures, will these figures be published? If the supplementary figures will not be published, it is necessary that part of the content of the supplementary material be included in the manuscript.

Reviewer 5 Report

Comments and Suggestions for Authors

The manuscript titled 'Computational modeling to identify drugs targeting metastatic castration-resistant prostate cancer characterized by heightened glycolysis' aims to identify drugs that can effectively treat castration-resistant prostate cancer. This form of cancer is notoriously difficult to treat, and the current range of available agents is limited. The study's findings are of great scientific and practical significance. The study of several drugs in vitro is a significant advantage of this work. This paper is suitable for printing in its current form. The topic is of great interest to practicing physicians and scientists. The paper deserves to be published in Pharmaceuticals.

Author Response

Thank you for the reviewer’s positive comment which is very encouraging for us. 

Round 2

Reviewer 1 Report

Comments and Suggestions for Authors

The authors have incorporated all the necessary modifications in the revised version of the manuscript.

Reviewer 4 Report

Comments and Suggestions for Authors

I have carefully reviewed the authors' responses to my submitted comments. The 19 comments that I submitted were correctly responded to by the authors, so I have no further comments or observations on the manuscript entitled "Computational modeling to identify drugs targeting metastatic castration-resistant prostate cancer characterized by heightened glycolysis" (Manuscript ID: pharmaceuticals-2965189).